# Enhancing Biosecurity in Mollusc Aquaculture: A Review of Current Isothermal Nucleic Acid Detection Methods

**DOI:** 10.3390/ani15111664

**Published:** 2025-06-04

**Authors:** Hoda Abbas, Gemma Zerna, Alexandra Knox, Danielle Ackerly, Jacinta Agius, Karla Helbig, Travis Beddoe

**Affiliations:** 1Department of Ecological, Plant and Animal Science, La Trobe University, Bundoora, VIC 3083, Australia; 19143727@students.latrobe.edu.au (H.A.); g.zerna@latrobe.edu.au (G.Z.); a.knox@latrobe.edu.au (A.K.); d.ackerly@latrobe.edu.au (D.A.); 2Department of Microbiology, Anatomy, Physiology & Pharmacology, La Trobe University, Bundoora, VIC 3083, Australia; j.agius@latrobe.edu.au (J.A.); k.helbig@latrobe.edu.au (K.H.); 3La Trobe Institute of Molecular Sciences, La Trobe University, Bundoora, VIC 3083, Australia

**Keywords:** molluscs, pathogens, isothermal amplification, point-of-care testing

## Abstract

The increase in farmed molluscs has heightened the risk of infection from various pathogens, potentially causing significant economic losses with limited treatment options. Consequently, there is a need for in-field diagnostics to enhance biosecurity management and prevent infections. This review provides an overview of the current molecular diagnostics for relevant diseases and modern isothermal techniques for nucleic acid detection, highlighting their application in point-of-care testing in the mollusc aquaculture industry.

## 1. Introduction

The human population is steadily increasing, with the global population recently surpassing eight billion people [1]. This growth escalates the demand for food production, which terrestrial-based meats cannot fulfil alone. Aquacultural products provide consumers with various protein sources, such as finfish, crustaceans, and molluscs. Commercial molluscs include bivalve molluscs (mussels, clams, scallops, cockles, and oysters), cephalopods (octopus, squid, and cuttlefish), and gastropods (abalone and conch) [2,3]. Molluscs represent 11% of the worldwide seafood trade, with over half of this comprising squids, cuttlefish, and octopus [3,4]. Molluscs are a good source of omega-3 fatty acids, iron, selenium, and zinc, while having lower levels of carbohydrates and fats compared to land-based proteins, making them a healthier dietary option than terrestrial meat [5,6]. Molluscs’ importance is not limited to their nutritional and commercial value, as they also play a critical role in maintaining the stability of the ocean’s ecosystem (Eutrophication) as they filter phytoplankton (mainly by bivalves) and help in carbonate buffering mechanisms [2,5]. Furthermore, the inedible shells of molluscs are used in medicine, cosmetics, jewellery, and in chemical industries such as dyes and catalysts [2,5,6,7].

Despite the global COVID-19 pandemic’s impact on international trade, the production of molluscs has continued to grow (Figure 1A). Since 1990, mollusc production (excluding cephalopods) has seen an annual growth rate of 2.7%, with a million ton increase in farmed molluscs from 2020 to 2022 (Figure 1A) [3,4]. In 2018, mollusc farming generated USD 34.6 billion in revenue, second only to finfish production, which brought in USD 139.7 billion [3]. Mollusc consumption reached an average of 3.1 kg per person in 2017 and still accounted for half of all shellfish consumption in 2022 [3,4]. In 2018, the production of shelled molluscs from marine and coastal aquaculture was 17.3 million tons (56.2%), which exceeded that of finfish and crustaceans, with production volumes of 7.3 million tons and 5.7 million tons, respectively [3].

Asia is the world’s largest producer of molluscs, with an annual production of 17,449,826 tons in 2022, far surpassing Europe (598,680 tons), followed by America, Oceania, and Africa [4]. Within Asia, China is the leading producer, generating approximately 16 million tons of highly commercial molluscs such as mussels, squid, cuttlefish, and abalone, with 75% of the world’s molluscs being farmed in 2022 [4] (Figure 1B).

The intensive expansion of aquatic farming, combined with poor environmental quality, nutrition, and farm management, makes cultivated mollusc species highly susceptible to disease. This vulnerability can lead to significant production losses when infectious disease outbreaks occur [8]. In a recent 2018 census study of aquaculture, performed by the Department of Agriculture in the USA, diseases were listed as the main cause of aquaculture production losses [3]. Molluscs are susceptible to many viral [9,10], bacterial [11], parasitic [12,13], and fungal diseases [14], causing mortality or slow growth rates that have detrimental impacts on the domestic and international markets. Identifying infectious pathogens quickly is crucial for sustaining the mollusc production industry.

## 2. Pathogens of High Biosecurity Concern in Global Mollusc Production and Their Current Molecular Diagnostic Methods

There have been significant outbreaks of various diseases on mollusc farms worldwide, some of which have resulted in 100% mortality rates. Most mollusc diseases are transmitted through direct contact between infected animals or by exposure to contaminated water [15]. Despite strict regulations on the trade of molluscs, infectious diseases continue to spread both among different species and within the same species globally [16,17,18,19]. The 2024 World Organization for Animal Health (WOAH) list of notifiable diseases includes five parasitic pathogens that have severe effects on molluscs globally: *Bonamia ostreae*, *Bonamia exitiosa*, *Marteilia refringens*, *Perkinsus marinus*, and *Perkinsus olseni*, along with abalone herpesvirus. Since clinical signs are not always present to confirm infection, diagnostic methods are essential to inform practices on controlling the spread of pathogens on farms and during the transportation of molluscs between countries. This summary outlines the current molecular detection methods used to identify mollusc diseases (Table 1).

### 2.1. Viral Diseases Affecting Molluscs

#### 2.1.1. Abalone Viral Ganglioneuritis (AVG)

The abalone production industry is under significant threat from the abalone herpesvirus (AbHV), also known as Haliotid herpesvirus-1 (HaHV-1) in Australia [20]. This neurotropic virus causes ganglioneuritis, leading to the condition termed abalone viral ganglioneuritis (AVG) [21]. The virus is believed to have first emerged in China in the late 1990s and later spread to Taiwan in 2003 [21]. The Taiwan outbreak mainly affected cultured (*Haliotis diversicolor supertexta*) or small abalone and resulted in losses for the industry exceeding USD 11.5 million [22,23,24]. In 2005, subsequent outbreaks in Victoria, Australia, led to 100% mortality in blacklip (*Haliotis rubra*), greenlip (*Haliotis laevigata*), and their hybrid due to AVG [25]. Abalone are generally vulnerable to AbHV at all life stages; however, complete resistance has been observed in the New Zealand paua *(Haliotis iris*) after artificial infection [26]. Additionally, disc abalone (*Haliotis discus hannai*) has been identified as an asymptomatic carrier of the virus, harbouring it without signs of disease under experimental conditions [20]. Further research is necessary to determine whether *H. discus hannai* has natural immunity or if environmental factors contribute to disease manifestation [20]. The AbHV virus is enveloped and icosahedral, measuring 100 nm, and is classified under the genus *Aurivirus* within the family *Malacoherpesviridae* and order *Herpesvirales* [27]. AVG is cytocidal, impacting the nervous system and causing tissue necrosis associated with the nerves [28]. Infected abalone during the Taiwan outbreak exhibited symptoms such as mantle recession and muscle atrophy, while Victorian abalone displayed symptoms like swollen mouths and prolapsed odontophores [24,28]. The variation in disease presentation among different abalone species complicates the diagnosis of pathogen infections based solely on clinical signs [21].

Due to the virus’s severity and rapid spread, prompt and accurate on-farm detection is crucial. However, the absence of a gold standard detection recommendation from the WOAH complicates matters. Corbeil et al. (2010) developed a TaqMan quantitative PCR (qPCR) assay capable of detecting fewer than 300 copies of a recombinant plasmid containing the target open reading frame (ORF) 38 (also known as 49) of the AbHV gene. The emergence of new AbHV-1 strains in Tasmania necessitated the development of new qPCR assays targeting ORFs 66 and 77 [15,29,30]. These new assays can detect all known strains of HaHV-1 in Australia, even during sub-clinical stages, but require extraction of the nerves, making them costly and time-consuming [29]. With the emergence of new variants and the high susceptibility of most abalone species to this virus, establishing a robust surveillance system, particularly in low-resourced laboratories, is essential. As there is currently no vaccine available against AbHV, quick and accurate diagnosis is vital for containing the virus before outbreaks occur.

#### 2.1.2. Abalone Shrivelling Syndrome (AbSS)

The inaugural documentation of abalone shrivelling syndrome (AbSS) dates back to 1999 in China, predominantly impacting small abalone [31,32]. Researchers identified a chimeric, double-stranded DNA virus in afflicted specimens, subsequently christened the abalone shrivelling syndrome-associated virus (AbSV). The genome of this viral pathogen shares extensive homology with both bacteriophages and bacteria, suggesting a complex ancestry [33]. After its initial emergence, AbSS has proliferated beyond China, reaching Taiwan and other nations, thereby affecting abalones across various developmental stages. The syndrome manifests through several clinical signs, including diminished appetite, pedal disc muscle wastage, mantle darkening, and a propensity for the abalones to dislodge from their reef habitats, often resulting in mortality post-detachment [34,35,36]. The most recent significant outbreak of AbSS was recorded in China in 2005. Despite the continued presence of the pathogen, there has been a noticeable decrease in its pathogenicity. This decline is potentially attributable to advancements in aquaculture practices, thereby enhancing survival rates. Additionally, observed mutations within the AbSV genome may have reduced the virus’s lethality [22,37].

To detect AbSS, nested PCR (nPCR) and qPCR assays have been developed, offering high levels of sensitivity and specificity. However, these methods require the dissection of animal mantles and feet for sampling, necessitating specialist expertise. Notably, the qPCR assay, which targets ORF2 of AbSS, can identify as few as 10 copies of the recombinant plasmid that harbours the target gene, highlighting the assay’s precision [34,35]. Despite this, molecular analyses of moribund abalones from Taiwan have shown a high genetic congruence with AbSV, yet in situ hybridisation (ISH) has not successfully linked these cases to AbSS, indicating a discrepancy in detection methodologies [31]. This disparity accentuates the imperative for developing accurate, swift, and easily deployable diagnostic tools capable of identifying the virus and its mutant strains.

#### 2.1.3. Acute Viral Necrobiotic Disease

The acute viral necrobiotic disease is caused by the acute viral necrobiotic virus (AVNV), which first appeared in China in the 1980s, leading to significant deaths among the zhikong scallop, *Chlamys farreri* [38]. AVNV is characterised by its spiked spherical envelope morphology containing DNA genomic materials, measuring approximately 170 nm in diameter [39]. The virus primarily targets the host’s epithelial cells and connective tissues, resulting in necrosis of essential organs such as the gills, mantles, intestine, and digestive gland [38,39]. The virus predominantly affects immature scallops around two years old, with infection rates peaking during the summer months when temperatures are between 25 and 27 °C. The mortality rate associated with AVNV infections is alarmingly high, estimated between 60 and 90%, and fatalities typically occur within 2–9 days post-infection [40,41].

Various techniques, including serological, microscopic, and molecular detection assays, have been developed to identify AVNV. Fluorescence quantitative PCR has demonstrated the ability to detect low numbers of the virus with high specificity [41,42,43,44]. However, despite advancements in rapid detection methods, none have been optimised for field application, as they primarily rely on aseptic isolation of gills and mantle tissue, which is challenging to perform on-site [43]. This highlights a critical gap in current diagnostic capabilities and emphasises the need for further research in this area.

#### 2.1.4. Infection with Ostreid Herpesvirus-1

The first documented case of ostreid herpesvirus-1 (OsHV-1) infection was identified in 1972 in the United States [16]. Despite extensive research, the definitive origin species responsible for transmitting OsHV-1 remains unknown. The virus has been detected in a variety of bivalve species, including oysters, mussels, clams, and scallops [16,21]. The virus’s ability to infect such a broad range of molluscs may be due to the co-cultivation of different aquatic organisms, which typically do not coexist, facilitating viral transmission. Another possibility is that OsHV-1 could be a mutation of another virus that originally infected a specific bivalve species before evolving to infect more broadly [17]. New variants of OsHV-1 continue to emerge, with the most recent being OsHV-1 µvar, which is characterised by a deletion of approximately 2.8 kbp from the original 167.8 kbp reference strain [45]. The µvar variant first appeared in France and is recognised as the most virulent form of the virus, devastating all life stages of the Pacific oyster in 2008, and subsequently spreading to Europe, Asia, New Zealand, and Australia, eventually being included on the WOAH list in 2013 [22,46,47]. Infected larvae often show reduced food intake and general activity, leading to increased mortality, while adult oysters display nonspecific symptoms like sluggish behaviour and gaping of the shell [33,48].

Efforts to isolate the virus through cell culture techniques have been unsuccessful, though infectivity has been established using filtrates from homogenised infected tissues [49]. Detection methods for OsHV-1 include qualitative and quantitative PCR, ISH, and immunochemistry techniques [48,50,51,52,53]. However, these methodologies are impractical for routine surveillance in the field due to their high cost, labour-intensive procedures, and reliance on specialised laboratory equipment and trained personnel for gonad and mantle dissection [50]. The WOAH recommends qualitative and quantitative PCR, followed by sequencing, for the confirmation of infections [33]. Given the lack of specific symptoms and the aggressive nature of OsHV-1, there is a pressing need for reliable, rapid, and cost-effective point-of-care detection methods.

### 2.2. Parasitic Diseases Affecting Molluscs

#### 2.2.1. Haplosporidiosis

The phylum *Haplosporidia* encompasses around 36 species of endoparasites that impact marine and freshwater invertebrates. This phylum includes *Urosporidium*, *Haplosporidium*, *Minchinia* taxa, and the recently added *Bonamia* [54]. A significant pathogen within this group is *Haplosporidium* spp., which leads to haplosporidiosis in Pacific oysters, *Crassostrea gigas*, across the USA, Europe, and Asia [55]. *Haplosporidium nelsoni*, commonly referred to as MSX, is the most extensively studied member of this genus. It was first identified in Delaware Bay in the 1950s, originally named *Minchinia nelson*, and is known for causing severe mortalities in eastern oysters, *C. virginica* [56]. *H. nelsoni* triggered a significant outbreak in oyster farms in the USA, culminating in 90% mortalities [57]. The impact of this outbreak led to the establishment of annual Oyster Mortality Conferences to investigate the issue throughout the following decade [58]. This disease is most prevalent in high-salinity waters, where the parasite damages the epithelial cells of the oysters’ digestive tubules, ultimately leading to their death. Notably, higher mortality rates are observed in the summer months, particularly when the water is at around 20 ppt salinity [59]. Another species that causes haplosporidiosis is *Haplosporidium costale* (SSO), which also targets eastern oysters and is morphologically similar to *H. nelsoni*.

Traditional morphological detection methods do not allow for differentiation between these two species, but certain molecular techniques, like conventional PCR (cPCR) and ISH, can identify *Haplosporidia* spp. Individually [60]. Penna et al. (2001) developed a multiplex PCR (mPCR) to detect three pathogens—*H. nelsoni*, *H. costale*, and *Perkinsus marinus*—for effective and rapid pathogen detection. However, despite the availability of numerous successful detection techniques, a quick and economical method that does not require trained specialists for tissue extraction and test execution is lacking [61,62].

#### 2.2.2. Bonamiosis

Bonamiosis, also known as Microcell disease, is caused by several members of the genus *Bonamia*. These small (2–3 µm) intracellular protozoa mainly infect haemocytes, but they can also be found in some extracellular tissues and may become systemic in severe cases [15,63,64]. Oysters are particularly vulnerable to this disease in waters that are cold and have high salinity. As a result, diseased oysters can experience mass mortality, with levels reaching 90%, which significantly impacts oyster populations and raises concerns about maintaining a balanced ecosystem, especially in Europe [65,66,67]. Currently, four members of *Bonamia* are known: *B. exitiosa*, *B. ostreae*, *B. roughley*, and *B. perspora*. *B. roughley* causes Australian winter disease, affecting Sydney rock oysters, *Saccostrea glomerata*, in southeast Australia [66]. The newly described *B. perspora* has been found in *Ostrea equestris* and *O. lurida* [66]. *B. ostreae* was first detected in the flat oyster species *O. edulis* in the 1970s in the USA, with serious consequences for the oyster industry in the northern hemisphere [65,68,69,70,71]. *B. exitiosa* was initially reported in New Zealand in 1985, causing severe damage to *O. chilensis* production, with a significant decrease in populations by 91% in 1990 [69,72,73,74]. Although oysters from the genus *Ostrea* were thought to be the only hosts for *Bonamia* spp., recent studies have shown that oysters from the genera *Crassostrea*, *Saccostrea*, and *Dendostrea* are also susceptible [65,67]. Infected hosts may show no symptoms or exhibit common sickness signs, such as ulcers, eroded or discoloured gills, and poor condition, which may lead to mortality [15,67]. Interestingly, some populations of *O. edulis* show better tolerance to the parasite, possibly due to low parasite levels or partial host resistance from long-term exposure [75]. Despite various attempts to clean and reuse previously infested waters to cultivate the oysters, the pathogen could not be eradicated [71,76]. Mortality peaks when infections coincide with *Marteilia refringens* or other *Bonamia* species. For example, dual infection with *B. ostreae* and *M. refringens* reduced flat oyster annual production in Europe from 29,595 tons to 5921 tons between 1961 and 2000, highlighting the need for specific detection methods to distinguish the causes of infection [65,72].

Various molecular biology techniques have been developed to detect *Bonamia* spp., including ISH, cPCR, qPCR, and mPCR; however, none of these methods are field-deployable, as they require gill tissue to be excised. This limitation poses significant challenges in real-time detection, especially in remote areas where immediate access to laboratory facilities is often not feasible [18,77,78,79]. Histology remains the gold standard for detection, but it is challenging to differentiate between *Bonamia* species using this method, especially in early infection stages [3]. Specific ISH assays can detect closely related *B. exitiosa*–*B. roughley* clade successfully [80]. The PCR-restriction fragment length polymorphism (RFLP) technique can differentiate polymorphisms among *Bonamia* species, yet it remains labour-intensive for field use [72,81]. A TaqMan qPCR assay designed to detect the internal transcribed spacer (ITS) sequence of *Bonamia* spp. Has demonstrated faster and more sensitive results than histology, and offers similar sensitivity to previously developed cPCR methods targeting the small subunit ribosomal DNA (SSU rDNA) [78,79]. Currently, the primary strategy for reducing infections involves cultivating lighter-weight oysters and minimising environmental stresses that could increase their susceptibility to disease, underscoring the urgent need for developing point-of-care tests for early detection [82].

#### 2.2.3. Marteiliosis

Marteiliosis poses a significant threat to molluscan populations, caused by protozoan parasites from the phylum *Cercozoa* and order *Paramyxida* [83]. This disease is linked to several species within the genus *Marteilia*, such as *M. sydneyi*, *M. granula*, *M. lenghei*, and others [84,85]. These parasites have led to considerable production declines in mussel, oyster, clam, and cockle farms globally [12,18,86,87,88]. Notably, marteiliosis can result in mass mortalities (50–90%) among adult oysters in their second year of infection [12,18,86,87,88]. *M. sydneyi* is particularly significant, primarily affecting the Sydney rock oyster [18,89,90,91]. The WOAH recognised this pathogen’s significance in the 1990s due to its effects on bivalve mollusc populations [89]. *M. refringens* was first discovered in the 1960s in flat oysters, and since then, infections have been reported in various life stages of other aquatic animals across Europe [18,90,92]. Recently, *M. refringens* has been classified into M and O types, specifically affecting mussels and oysters [18,90,93]. While the lifecycle of *Marteilia* spp. remains partly unclear, it has been established that the parasite follows an indirect life cycle as other marine organisms are involved. However, the parasite can survive up to three weeks outside an animal’s body in suitable conditions [92,94]. The pathology of marteiliosis differs with the specific pathogen and host, usually leading to reduced growth rates, poor health, tissue necrosis, and body shrinkage in infected molluscs [94,95,96,97,98]. In advanced stages, the accumulation of parasites in the digestive gland can lead to starvation and death [94,95,96,97,98]. Efforts to breed more resistant oyster species have faced challenges due to the wide susceptibility among various species; however, the Pacific oyster shows some resistance [12].

Detection and identification of *Marteilia* species have traditionally relied on histological methods, but molecular techniques are now more effective for species-level identification. The gold standard for detecting marteiliosis remains histology, often confirmed with ISH assays to validate findings [99]. PCR techniques, including generic PCR and nPCR assays, have been developed to detect various *Marteilia* species and differentiate between types, providing quicker identification methods [90,93]. Recently, mPCR assays have been introduced to detect pathogens from both *Marteilia* and *Bonamia* genera; however, these methods still require invasive tissue extraction, which is not ideal for field detection [18].

#### 2.2.4. Marteilioides

Marteilioides, a disease first identified in the 1970s within a Korean farming operation of Pacific oysters, poses a significant threat to aquaculture due to its pathogenic effects on oyster populations [67,100]. Infected specimens initially displayed yellowish, spherical nodules on the mantle, which were mistakenly thought to be amoebic infections [100]. However, further investigations using transmission electron microscopy (TEM) clarified this misidentification, leading to the classification of the new genus *Marteilioides* [100,101]. Among the species in this genus, *Marteilioides chungmuensis* has particularly harmful effects on the oysters’ reproductive systems, targeting the oocyte’s cytoplasm, obstructing egg release from the ovarian follicle, and causing infertility. This then led to a significant decline in seed oyster production, adversely impacting the aquaculture industries in Korea and Japan [67,102,103]. Additionally, *M. branchialis* has been linked to infections in the gills of Sydney rock oysters, with pathogenic severity increasing when co-infected with *M. sydneyi*, resulting in higher mortality rates [104].

A study assessing the sensitivity and specificity of current detection methods for *Marteilioides* spp. Found that ISH showed greater sensitivity compared to histology alone [105]. The ISH assay identified several immature parasites in the early development stages. Although PCR demonstrated high sensitivity, inconsistencies with histology and ISH led to many samples being incorrectly classified as false positives or negatives [105]. Despite ISH’s superior performance over histology and PCR, its labour-intensive and time-consuming nature, along with the requirement for specialists to aseptically dissect the animal for sampling, remains a significant disadvantage. These factors limit its practicality as a point-of-care detection method, particularly in urgent situations where rapid diagnosis is essential.

#### 2.2.5. Denman Island Disease

Denman Island recorded the first observed case of Denman Island disease in 1960, which primarily occurs when temperatures fall below 10 °C [67]. The causative agent is the intracellular protistan parasite, *Mikrocytos mackini*. There is limited information regarding its morphology, genomic DNA, and host interactions [106]. Denman Island disease affects various oyster species across multiple countries [66,74]. Older oysters are generally more severely affected; symptoms include haemocyte infiltration and necrosis, leading to mortality rates of approximately 30% [67]. The disease was previously on the WOAH notifiable diseases list and is currently included as an exotic disease in European (EU) legislation updated in 2018 [66]. A particularly intriguing characteristic of *M. mackini* is its lack of mitochondria, complicating efforts to determine its evolutionary position [106]. Further genetic studies have indicated that *M. mackini* has evolutionarily reduced mitochondrial-related organelles (MROs) known as mitosomes, which replace traditional mitochondria [106].

Significant advancements in detecting this parasite have been made with the development of sensitive and specific assays, including cPCR and fluorescent in situ hybridisation (FISH), targeting the SSU rDNA of *M. mackini* [107]. Additionally, an undefined qPCR assay was created to target the ITS-2 region in the rDNA of *M. mackini*. This assay can detect as few as 2–5 copies of genomic DNA from samples collected from the mid-body cross-section of oysters, which is notably more effective than the poorer DNA extraction results obtained from the mantle or adductor muscle [108]. However, the variability in results from samples taken from different parts of the oyster highlights the challenges involved in this process. This inconsistency underscores the need for expertise in sampling techniques to ensure accurate and reliable detection, as different anatomical regions may yield DNA of different quality and quantity.

#### 2.2.6. Perkinsosis

The genus *Perkinsus* comprises various species that infect hosts through direct contact, causing a condition known as perkinsosis. This disease can lead to numerous detrimental effects on host organisms, such as severely retarded growth, behavioural changes, inflammation, necrosis, and a significant decline in physiological functions like growth, gonadal maturation, reproduction, and immunocompetence. These adverse effects can lead to mortality rates as high as 95% after a year of infection, particularly when water temperatures rise above 20 °C [67,109,110,111,112]. Several *Perkinsus* species have been documented, including *P. atlanticus*, *P. qugwadi*, *P. andrewsi*, *P. chesapeaki*, *P. mediterraneus*, *P. honshuensis*, *P. beihaiensis*, *P. marinus*, and *P. olseni* [112,113,114]. However, only a subset of these species meets the recognised criteria for classification within the genus. For example, *P. andrewsi* and *P. atlanticus* have been identified as synonymous with *P. chesapeaki* and *P. olseni*, respectively. Conversely, *P. qugwadi* exhibits distinct phenotypic traits that do not align with established diagnostic markers of the genus, such as its reaction to Ray’s fluid thioglycolate medium (RFTM) [115]. *P. marinus* was first observed in Mexico during the 1940s in eastern oysters, causing perkinsosis disease in some oysters and clams in Europe and North America [116,117]. *P. olseni* was first reported in Australia in 1972, affecting the blacklip abalone. A recent outbreak of *P. olseni* in Australia led to a significant drop in abalone production from 300 tons to 94 tons in 2011, causing an annual loss of 500,000 AUD for abalone producers in South Australia since its emergence [113]. The expansive diversity of the *Perkinsus* genus increases the range of molluscs susceptible to perkinsosis, including oysters, mussels, cockles, clams, and abalone. This broad host range has serious ecological and economic implications, resulting in increased mortality rates in mollusc aquaculture and a decline in the market value of affected species [114,118]. Continued research and documentation of *Perkinsus* species emphasise the need for ongoing surveillance and mitigation strategies to protect mollusc populations and the aquaculture industry.

The WOAH recommends starting with genus-specific PCR before employing species-specific detection assays for positive samples [15]. Recent advancements in molecular diagnostics have led to the development of genus-specific qualitative and quantitative PCR assays targeting various *Perkinsus* genes, including ITS, actin, and the ribosomal RNA large subunit (rRNA LSU) gene [119,120,121,122]. A notable innovation is a universal PCR assay that detects a 703 bp sequence unique to the genus *Perkinsus*, with the exception of *P. qugwadi* [123]. Given the small size of these parasites, standard histological methods are insufficient for accurate diagnosis, prompting the development of a universal ISH assay targeting the rRNA SSU domain specific to the genus *Perkinsus* [124]. Species-specific ISH assays have also been created for *P. beihaiensis*, *P. chesapeaki*, *P. honshuensis*, and *P. mediterraneus* [122,125,126]. In addition, species-specific PCR assays have been designed to accurately identify *P. honshuensis*, *P. chesapeaki*, and *P. beihaiensis*, reflecting ongoing efforts to improve detection methods [122,127,128]. A multiplex PCR-enzyme-linked immunosorbent assay (ELISA) has been developed to identify the intergenic spacer (IGS) sequence of *P. marinus*, *P. atlanticus*, and *Perkinsus* spp., demonstrating sensitivity to as low as 1 pg of DNA, which is 100 times more sensitive than cPCR [129]. Furthermore, a highly sensitive and species-specific qPCR assay for detecting the ITS sequence of *P. marinus* has been developed, able to detect the pathogen in environmental water samples [123]. However, these remain laboratory-based techniques, with no in-field options available. Species-specific PCR primers can amplify a 455 bp region of the ITS specific to *P. olseni*’s rRNA gene complex, offering exceptional sensitivity to detect as few as one pathogen cell in 30 mg of tissue however, genomic DNA extraction from excised mantle and gill tissues is a labour-intensive process that requires specialised techniques and expertise, which may not be readily available in farm settings for point-of-care detection [130]. Through these advancements, the landscape of *Perkinsus* spp. diagnostics continues to evolve, providing refined tools for rapidly detecting this parasite, which is crucial for effective management and control strategies in affected marine environments.

**Table 1 animals-15-01664-t001:** Pathogens affecting commercial molluscs and their susceptible species and current molecular detection methods.

Pathogen	Susceptible Mollusc (s)	Detection Method
**Virus**		
**Abalone herpesvirus (AbHV)**	Blacklip abalone ^1^ Brown abalone ^2^ Disc abalone ^2^ Greenlip abalone ^1^ Pink abalone ^2^ Small abalone ^1^ Tiger abalone ^1^	cPCR [131] Sequencing [131] qPCR [29,30] ISH [132]
**Abalone shrivelling syndrome (ASSV)**	Disc abalone Small abalone	qPCR [35] nPCR [34]
**Acute Viral Necrobiotic Virus (AVNV)**	Scallops	cPCR [41] qPCR [41,133,134]
**Ostreid herpesvirus-1 (OsHV-1)**	Ark clams Australian flat oyster Bay scallops Blood clam Blue mussels Chilean oyster European clam Flat oyster Great scallop Hairy mussels Manila clam Pacific oyster Portuguese oyster Sydney cockle Sydney rock oysters Telline Virescent oyster Whelks	PCR [48] ISH [51,52] qPCR [53]
**Parasite**		
***Bonamia* spp.**	Australian flat oyster ^3^ Chilean oyster ^1^ Crested oyster ^1^ Dwarf oyster ^1^ European flat oyster ^1^ Hawaiian oyster ^1^ Jinjiang oysters ^2^ Olympia oyster Pacific oyster ^1^ Portuguese oyster ^1^ Suminoe oyster ^1^ Sydney rock oysters	cPCR [77,78] qPCR [79] mPCR [18] ISH [78,80]
** *Bonamia exitiosa* **	Argentinian flat oyster Australian flat oyster ^1^ Chilean oyster ^1^ Dwarf oyster ^1^ Eastern oyster European flat oyster ^1^ Olympia oyster Pacific oyster Sydney rock oyster	qPCR [69] cPCR and sequencing [78,135,136] PCR-RFLP [72] mPCR [69] ISH [78,80,137]
** *Bonamia ostreae* **	Argentinian flat oyster ^3^ Asiatic oyster Australian flat oyster Chilean oyster European flat oyster ^1^ Pacific oyster ^3^ Portuguese oyster Suminoe oyster ^1^	ISH [78,107] qPCR [79,138,139] cPCR [77,78,140] mPCR [69] PCR-RFLP [72]
***Haplosporidium* spp.**	Australian flat oyster Blue mussel California mussel Cockles Eastern oyster European flat oyster Freshwater snails	qPCR [141,142] cPCR [60,143,144] mPCR [62] ISH [60]
** *Haplosporidium nelson* **	Eastern oyster Pacific oyster	ISH [145] cPCR [146] qPCR [147] mPCR [62]
***Marteilia* spp.**	Argentinian flat oyster ^3^ Australian flat oyster Banded Carpet Shell Blacklip oyster Blacklip pearl oyster Blue mussel Calico scallop Chilean oyster Common cockle Dwarf oyster Eastern oyster European flat oyster Grooved razor clam Hooded oyster Iwagaki oyster Jackknife clam Manila clam Maxima clam Mediterranean mussel Northern horse mussel Pacific oyster Palourde clam Peppery furrow shell Pod razor Puelchean oyster Pullet carpet shell Rock oyster Striped venus clam Suminoe oyster Venerid clam	cPCR [89,90,93] ISH [93] RFLP-PCR [148]
** *Marteilia refringens* **	Argentinian flat oyster ^2^ Asiatic oyster ^1^ Australian flat oyster ^2^ Banded Carpet Shell Blue mussel ^1^ Calico scallop ^2^ Chilean oyster ^1^ Common cockle ^1^ Dwarf oyster ^2^ Eastern oyster ^1^ European flat oyster ^1^ Grooved razor clam ^1^ Hooded oyster ^1^ Jackknife clam Mediterranean mussel ^1^ Olympia oyster ^1^ Pacific oyster ^2^ Palourde clam Planktonic copepods ^2^ Pod razor Pullet carpet shell Small brown mussel ^2^ Striped venus clam ^1^	nPCR [90,149] cPCR and sequencing [89,93,97] mPCR [18] qPCR [150] ISH [89,90,99,151]
** *Marteilia sydneyi* **	Flat oyster Sydney rock oyster	cPCR [152] mPCR [18] ISH [153]
***Marteilioides* spp.**	Manila clam Northern blacklip oyster Pacific oyster Suminoe oyster	nPCR [154]
** *Marteilioides chungmuensis* **	Iwagaki oyster Manila clam Pacific oyster Pacific oyster Suminoe oyster	cPCR [102,155] ISH [102]
** *Mikrocytos mackini* **	Eastern oyster European flat oyster Olympia flat oyster Pacific oyster	cPCR [156] qPCR [108] ISH [157] FISH [156]
***Perkinsus* spp.**	Asian littleneck clam Baltic clam Eastern oyster European flat oyster Hong Kong oyster Mangrove oyster Manila clam Palourde clam Soft shell clam Stout tagelus Suminoe oyster Sydney cockle ^1^ Yesso scallop	cPCR [122,158] ISH [124] PCR—DGGE ^1^ [159] mPCR-ELISA [129]
** *Perkinsus andrewsi* **	Baltic clam	cPCR [127]
** *Perkinsus atlanticus* **	Palourde clam	cPCR [160] mPCR-ELISA [129]
** *Perkinsosis marinus* **	Baltic macoma Blue mussel Cortez oyster ^1^ Eastern oyster ^1^ Mangrove oyster ^1^ Pacific oyster ^1^ Soft-shelled clam Suminoe oyster ^1^	cPCR [123,158] ISH [124,126,161] qPCR [120,123] mPCR-ELISA [129] RFLP-PCR [162]
** *Perkinsosis olseni* **	Akoya pearl oyster ^1^ Asian littleneck clam ^1^ Australian flat oyster ^1^ Blacklip abalone ^1^ Blacklip pearl oyster ^1^ Crocus clam ^1^ European aurora venus clam ^1^ Giant clam ^1^ Greenlip abalone ^1^ Green-lipped mussel ^1^ Japanese pearl oyster ^1^ Kumamoto oyster Manila clam ^1^ Maxima clam ^1^ New Zealand ark shell ^1^ New Zealand cockle ^1^ New Zealand pauaa ^1^ New Zealand pipia ^1^ New Zealand scallop ^1^ Pacific oyster ^1^ Pearl oyster ^1^ Pullet carpet shell ^1^ Sand cockle Silverlip pearl oyster ^1^ Staircase abalone ^1^ Suminoe oyster ^1^ Sydney cockle ^1^ Venerid clam ^1^ Venerid commercial clam Venus clam Wedge shell Whirling abalone ^1^	ISH [124,161,163] cPCR [123,161] qPCR [123]

^1^ Naturally susceptible, ^2^ Experimentally susceptible, ^3^ No complete evidence of susceptibility, in situ hybridisation (ISH), fluorescent ISH (FISH), polymerase chain reaction (PCR), conventional PCR (cPCR), nested PCR (nPCR), quantitative PCR (qPCR), multiplex PCR (mPCR), enzyme-linked immunosorbent assay (ELISA), restriction fragment length polymorphism (RFLP), denaturing gradient gel electrophoresis (DGGE).

## 3. Isothermal Nucleic Acid Detection Methods

There is a need to develop techniques for the early identification of pathogens in farmed molluscs to allow for the decision-making process for controlling the infection, isolating, and treating the infected members, which can decrease the chances of mass outbreaks. Improved molecular technology for the detection of pathogen nucleic acid, such as polymerase chain reaction (PCR), quantitative PCR (qPCR), and ISH, has been developed for a range of diseases affecting molluscs (Table 1). However, these require specialised laboratories and trained personnel [79,164,165,166].

Field-deployable qPCR instruments are emerging, particularly for biodefence [167], but their use is still nascent. The Genesig q16 (Primerdesign, Chandler’s Ford, UK) is portable and field-operable, validated for salmonid alphavirus detection [168], with over 500 assays claimed (https://www.primerdesign.co.uk/products/instrumentation/genesig-q16-real-time-pcr-instrument/ accessed on 29 May 2025). Similarly, BioFire FilmArray (BioFire Diagnostics; https://www.biofiredx.com/ accessed on 4 May 2025) is a compact, portable qPCR system for anthrax and other biodefence threats, functioning as a closed system for nucleic acid extraction and amplification, albeit processing one sample at a time. In addition, both the Genesig and BioFire instruments require alternating current (AC) power to operate, which could limit their usefulness in the field.

In recent years, there has been development of several isothermal nucleic acid amplification technologies, which have the advantages of being simple to use, low cost, field deployability, and rapid results [19,169,170,171,172]. This section outlines the isothermal amplification techniques (Figure 2) that have been applied to mollusc viral and parasitic disease diagnostics to fill the gap between quick diagnostics and accuracy.

### 3.1. Loop Mediated Isothermal Amplification (LAMP)

Loop-mediated isothermal amplification (LAMP) is one of the most widely used isothermal methods for detecting pathogens affecting molluscs (Table 2). The first LAMP assay was developed by Notomi et al. in 2000 to detect the hepatitis B surface antigen (HBs) region of the human hepatitis B virus. LAMP relies on an auto-cycling strand displacement mechanism for DNA synthesis. In this method, four to six primers are designed to target six to eight template regions in the presence of an isothermal DNA polymerase (Bst) [173,174]. The four main primers include two outer primers, F3 and B3, which function similarly to the forward and reverse primers in PCR. Additionally, two inner primers, the forward inner primer (FIP) and the backward inner primer (BIP), are designed to complement two distinct regions on the sense and antisense strands of the target sequence. The FIP is composed of F2 and F1c, which are complementary to F2c on the sense strand and F1 on the antisense strand, respectively. The BIP is formed by connecting B2 and B1c, which are complementary to B2c on the antisense strand and B1 on the sense strand, respectively (Figure 2(A2)). The reaction can be started by either FIP or BIP, although here we will start with FIP for easier explanation. The reaction begins when F2 of the FIP hybridises with its complementary region F2c on the target strand. The F3 primer binds to an external region of the target and elongates, displacing the newly formed strand produced by FIP. The displaced strand, now containing the F1c region, anneals to its complementary F1 region on the new strand, forming a loop structure at one end. This new strand serves as a template for BIP. The B3 outer primer then displaces the new product formed by BIP, resulting in a dumbbell-shaped structure. The loop region acts as a template for the forward loop primer (FLP) and backward loop primer (BLP), providing additional amplification starting points. FIP, BIP, FLP, and BLP continue to prime on the loop and dumbbell-shaped DNA strands, leading to the formation of a cauliflower-like structure where multiple amplification, elongation, and displacement events occur simultaneously [173]. LAMP typically targets DNA samples or RNA after converting it to complementary DNA (cDNA). LAMP does not need extensive sample preparation and is more tolerant to inhibitors than PCR [173,175]. The LAMP steps occur at a constant incubation temperature of 60–65 °C for 15–65 min, which eliminates the need for complicated instruments like thermocyclers. Instead, the process can be easily performed using a water bath or heat block. LAMP products can be visualised using various methods, including turbidity, colourimetry, electrochemical detection, agarose gel electrophoresis (AGE), or real-time detection with fluorescence [176,177,178,179].

**Figure 2 animals-15-01664-f002:**
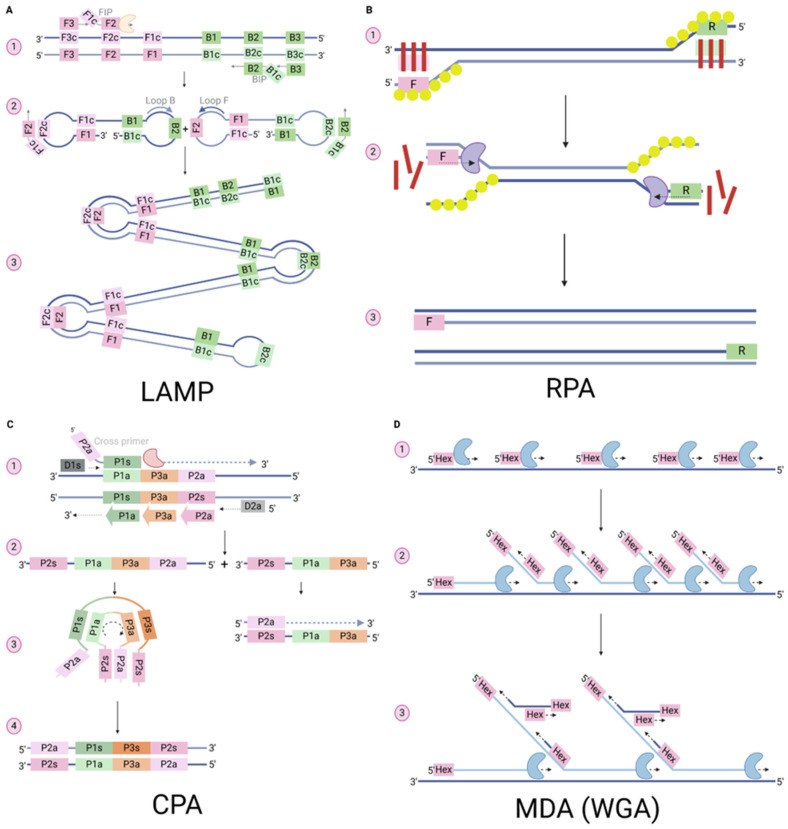
Comparison between the amplification mechanism for loop-mediated isothermal amplification (LAMP), recombinase polymerase amplification (RPA), cross-priming amplification (CPA), and multiple displacement amplification (MDA). (**A**) LAMP. (**A1**) The annealing of inner primers (FIP/BIP) to the target strand and elongation with the isothermal polymerase (orange blob), then displacement of the new strands by the outer primers (F3/B3). (**A2**) The newly amplified strands with joined F1c and B1c annealed to their complementary regions on the same strand, forming dumbbell-shaped DNA. (**A3**) Multiple amplifications and strand displacements by inner and loop primers, resulting in loop and stem-like structures. (**B**) RPA. (**B1**) Recombinase protein (red strips)-primer complex with each of the forward (pink) or reverse (green) primer hybridises to the target sequence on sense and antisense strands, respectively, with the now free single-stranded DNA stabilised by single-stranded binding proteins (yellow circles). (**B2**) Following complex disassembly, elongation of the new strand is initiated by polymerase DNA (purple blob). (**B3**) The reaction continues with the new double strands serving as templates for the recombinase-primer complex. (**C**) CPA. (**C1**) Half of the cross primer anneals to the complementary region on the target strand, then extension occurs using the polymerase (light red). The generated strand with flanking sequence to the 5′ end allows it to be displaced by the outer forward primer (dark grey). (**C2**) Long and short products are generated. The longer amplification products have the cross-priming complementary region (pink) to another sequence on the same strand. (**C3**) The formation of the hairpin structure from the annealing of the complementary region of the cross primer and the complementary sequence on the same strand, while the short formed strand cannot produce the hairpin structure. (**C4**) Newly formed hairpin structures serve as templates for primers and produce more amplicons. (**D**) Whole genomic amplification (WGA)-MDA. (**D1**) Many random hexamer primers (pink) hybridise to the template and extend using phi29 DNA polymerase (blue blob). (**D2**) Occurrence of strand displacement by the new extending primers. (**D3**) Continuous amplification of the newly created strands results in the formation of networks of branched DNA structures. Created in BioRender. Abbas, H. (2025) https://BioRender.com/w75z807 (accessed on 3 February 2025).

### 3.2. Recombinase Polymerase Amplification (RPA)

Recombinase polymerase amplification (RPA) is another successful isothermal nucleic acid amplification technique, first developed by Piepenburg et al. in 2006 [169]. This method utilises cellular DNA proteins with functions in synthesis, repair, and recombination. RPA requires only two primers (forward and reverse), a recombinase protein derived from the T4 bacteriophage, a high molecular weight polyethylene glycol acting as a crowding agent, a single-stranded DNA binding protein, strand-displacing DNA polymerase, nucleotides, and ATP. An optional probe can be included to increase the specificity of the assay [169,180,181,182]. The recombinase protein binds to one of the primers, forming a recombinase-primer complex in the presence of ATP and the crowding agent (Figure 2(B2)). This complex scans the double-stranded DNA until it locates a sequence complementary to the primer. The primer complex then invades the double-stranded DNA and hybridises with the target region, leaving the complementary strand exposed and seeking to reunite with the original complementary strand. The single-strand binding protein stabilises the exposed strand, preventing it from re-annealing. Once the complex disassembles, the DNA polymerase attaches to the 3′ end of the primer and elongates it. This cycle repeats until the end of the incubation time [169]. The optimal temperature for RPA amplification ranges between 37 and 42 °C, and the process does not require a denaturation step, which differs from cPCR [180,181]. DNA or RNA amplification can be completed within 5–20 min, and the resulting amplicons can be observed by many tools, including AGE or lateral flow devices (LFD) [181,183]. RPA is currently commercialised by TwistDx, highlighting its ease of use by untrained personnel and its high inhibition tolerance, allowing for the rapid processing of various field samples within minutes [184].

### 3.3. Cross-Priming Isothermal Amplification (CPA)

Cross-priming isothermal amplification (CPA) is a recently developed isothermal amplification method used to detect pathogens affecting molluscs. CPA was developed by Ustar Biotechnologies Co., Ltd. and utilises 5–8 primers and probes, with at least one of them being a cross-linked primer [185,186,187]. CPA relies on the use of at least one cross primer, which consists of two connected sequences: one complementary to the template, and the second designed as a flanking sequence at the 5′ end that allows for strand displacement. This flanking sequence is complementary to a region on the newly amplified strand, enabling the formation of a hairpin structure that serves as a template for further amplification by other primers [187]. Reverse primers are designed to anneal to the antisense strand in tandem, providing a region for the nicked double-stranded DNA to elongate with the help of the isothermal strand-displacing DNA polymerase (Bst) [185]. The reaction is initiated when the forward cross primer anneals to the sense strand and extends. An outer forward primer then anneals to a region upstream of the cross primer, displacing the newly synthesised strand (Figure 2(C2)). This newly displaced strand, with the 5′ end attachment, serves as a template for the reverse primers. The displacement of the reverse-primed strands results in two different amplicons: a shorter one that lacks the cross-linkage complementary sequence, and a longer one that contains the complementary region and forms a hairpin structure. These latter structures continue to act as templates for other primers, producing more of the previously described short and long amplicons, and the amplification continues until the reaction concludes [185]. CPA can amplify both DNA and RNA samples at temperatures ranging from 60 to 68 °C without the need for an initial denaturation step [185,188]. The average amplification time is 40–60 min, and the products can be visualised using turbidity, colourimetry, or AGE [180,186,189,190].

### 3.4. Multiple Displacement Amplification (MDA)

Multiple displacement amplification (MDA) is a widely recognised isothermal amplification technique developed by Dean et al. in 2002. Initially called multiply-primed rolling circle amplification, and was originally designed to amplify plasmids. The technique relies on the use of random or partially random hexamer primers for strand-displacement and amplification facilitated by the phi29 DNA polymerase. This enzyme is particularly suited for MDA due to its high strand-displacement capabilities, allowing it to synthesise up to 70,000 nucleotides in a single reaction. Additionally, phi29 DNA polymerase boasts exceptional fidelity and exonuclease resistance under isothermal conditions [191,192]. One of MDA’s most notable applications is whole genome amplification (WGA). In WGA-MDA, random hexamer primers anneal to multiple sites on the template DNA, and phi29 DNA polymerase extends these primers across the entire template (Figure 2(D2)). As new strands displace the original complementary strands, they themselves become templates for further amplification. This reaction produces large amounts of highly branched DNA [191]. Unlike other isothermal amplification techniques such as LAMP, RPA, and CPA, MDA requires an initial denaturation step at 94 °C for several minutes, followed by incubation with the enzyme at 30 °C for several hours. The reaction is terminated by deactivating the enzyme at 65 °C [193,194]. The efficiency of the assay can be improved by adding exonuclease-resistant primers to the 3′ end of the template or by using crowding agents [191]. The success of amplification can be visualised using AGE [195]. WGA-MDA can amplify DNA and RNA samples and is especially useful for amplifying very small quantities of genetic material. It can generate approximately 20 µg of DNA from a single genomic DNA copy, which is invaluable for samples that cannot be enriched or cultured [19,192,196,197]. However, WGA-MDA is highly sensitive to contaminants because of its lack of specificity (random amplification) and must be performed under strictly sterile conditions [195,198].

## 4. Application of Isothermal Amplification of Viral Pathogens Infecting Molluscs

The use of isothermal amplification has seen a remarkable increase over the last decade, offering a valuable option for pathogen detection directly in the field. This technology enables faster decision-making and more effective infection control, significantly enhancing the ability to respond to outbreaks and manage health risks in farmed molluscs. In 2014, four LAMP primers were designed to target a sequence of the DNA polymerase gene, which is specific to AbHV. The assay was performed at 63 °C for an hour and could detect as low as 100 virus copies/µL in nerve tissues extracted from moribund abalone. The results could be observed by AGE as well as the naked eye by using UV light to visualise the fluorescent dye [176]. Furthermore, a real-time RPA assay was also designed to detect AbHV. The assay was faster and more sensitive than the corresponding PCR tests [170]. Genomic DNA was extracted from the muscle tissue of infected *H. diversicolor*, and specific primers and a probe were designed to amplify ORF49 (also called ORF38) of AbHV. The AbHV-RPA assay could detect 100 copies/reaction in 20 min at 37 °C, with no cross-reactivity with any of the closely related pathogens. The results could be observed in real-time, as well as with the AGE technique [170]. Although both isothermal assays demonstrate high sensitivity, the TaqMan PCR assay recommended by WOAH remains superior, with a detection limit as low as 30 copies per reaction. To enhance the performance of current isothermal detection methods, optimisation of sampling and nucleic acid extraction protocols is necessary. At present, both assays rely on abalone tissue processed using commercial extraction kits, which are costly, impractical for field use, and require trained personnel [15,30]. Therefore, in-field validation of both isothermal techniques under realistic aquaculture conditions is essential to support their future application in routine diagnostics.

To safeguard abalone and scallops from pathogenic viral infections, various LAMP assays have been developed as rapid and sensitive diagnostic tools. A LAMP assay targeting the AVNV was optimised for detection in scallop kidney tissues. Using four LAMP primers and a water bath as the heat source, the assay demonstrated a detection sensitivity of approximately 1 fg of genomic DNA—twice as sensitive as cPCR. Amplification results were visually confirmed with GeneFinder™ dye after one hour of incubation [199]. Although the assay can sensitively detect low levels of the virus in animal tissue, its dependence on invasive sampling methods limits its practicality for field application. To enhance its suitability for in-field diagnostics, non-invasive sampling approaches should be developed and thoroughly evaluated for the detection of AVNV.

Additionally, a LAMP assay targeting the ORF2 sequence of AbSV achieved a detection limit of just 10 copies of AbSV vectors within one hour at 60 °C, with amplification visualised using SYBR Green I dye and AGE [200]. The detection limit of this assay is comparable to that of the corresponding qPCR; however, both currently require invasive tissue sampling through animal dissection. Incorporating non-invasive sampling methods would enhance their field deployability, and the inclusion of loop primers could further accelerate reaction kinetics, reducing amplification time and improving overall diagnostic efficiency.

Various isothermal diagnostic tools have been developed to enhance the efficiency of OsHV-1 detection. A LAMP assay targeting the ATPase subunit of the OsHV-1 DNA-packaging terminase gene, encoded by ORF109, demonstrated a detection limit as low as 20 copies. Using four primers, the assay successfully detected the virus within one hour at 60 °C in oyster tissues. However, the detection dye had to be added post-amplification, posing a risk of cross-contamination [201]. Later, an attempt for in-field use without additional handling, termed single tube LAMP assay, was developed to detect the ORF4 sequence of OsHV-1. One-step reaction was performed by adding hydroxynaphthol blue (HNB) dye to the reaction mix to avoid potential contamination from the aforementioned LAMP assay [202]. Although the duration of this assay was reduced to 10 min instead of one hour by adding a helicase enzyme, further improvements are still needed to enhance its sensitivity, which is currently around 1000 copies [202].

Furthermore, a real-time RPA assay was designed to target ORF95 of OsHV-1 in genomic DNA samples extracted from ark clams [203]. In 2020, the same primers and probe from this assay were utilised to develop an in-field isothermal detection method by combining RPA with electrochemical detection. This new assay took 20 min to amplify a minimum of 207 copies of OsHV-1 at room temperature, whereas the original RPA assay was capable of detecting as low as five copies within the same time frame [203,204]. A CPA assay was also designed to specifically amplify the variant SB strain of OsHV-1, which was associated with mass mortalities in blood clam broodstocks [205]. Primers were generated targeting the conserved sequence of the OsHV-1-SB strain. The reaction was performed for an hour at 63 °C and could detect as few as 30 copies/µL of the positive control plasmid. The results can be observed after simple centrifugation or by using GeneFinder^TM^ dye [171]. Both the developed isothermal assays and the conventional cPCR and qPCR methods recommended by the WOAH currently rely on combined gill and mantle tissues as the sample source [15]. For more efficient diagnostics and routine surveillance, it is recommended to evaluate simple, non-invasive sampling techniques—such as swabbing—as alternative sources of DNA for both isothermal and PCR-based assays. Implementing such methods would improve field applicability, particularly in farm environments and resource-limited laboratories, while also supporting mollusc health monitoring and minimising the need for unnecessary animal sacrifice.

## 5. Application of Isothermal Amplification of Parasitic Pathogens Infecting Molluscs

Isothermal amplification, particularly loop-mediated isothermal amplification (LAMP), has shown significant promise in the field of mollusc disease diagnostics, providing a means to overcome the limitations of traditional detection methods. For instance, *Bonamia* species, such as *B. exitiosa* and *B. ostreae*, are unculturable protozoa, which presents a challenge for their detection and diagnosis [19]. To address this, three LAMP assays have been developed and evaluated to detect *B. exitiosa*, *B. ostreae,* and *Bonamia* genus members [206]. The assays were tested using portable real-time Geni devices and a real-time thermocycler. Each of the developed assays employed six LAMP primers individually targeting unique regions of *Actin*, *Actin-1,* and *18S* specific to *B. exitiosa*, *B. ostreae,* and the *Bonamia* genus, respectively. The *B. exitiosa* and *B. ostreae* LAMP assays were species-specific with a limit of detection of 50 copies/µL in a 30 min reaction time, which was only 10-fold less sensitive than the qPCR. Notably, the *B. exitiosa* LAMP assay did not cross-react with any of the non-target samples; the *B. ostreae* LAMP assay detected *Mikrocytos veneroïdes* after 30 min and, therefore, was deemed negative. The generic assay detected as few as 10 copies/µL, but the amplification was unreliable, as successful amplification occurred in only five out of ten runs. Therefore, the limit of detection was set to be 50 copies/µL, which is similar to the corresponding qPCR assay. Furthermore, the generic LAMP assay exhibited cross-reactivity with samples highly infected with *Haplosporidium costale* after 26 min, with a similar melting temperature as the target [206]. This latent amplification of *Haplosporidium costale* necessitates confirmatory testing in regions where both pathogens may be present. While isothermal detection methods offer faster results and require less technical expertise compared to traditional approaches such as tissue imprints, histology, and the WOAH-recommended PCR assays, all of these methods still depend on invasive sampling and complex DNA extraction protocols. To facilitate regular surveillance under field conditions, there is a critical need to develop non-lethal sampling techniques and simplified extraction methods.

Many innovative methodologies are being developed to overcome the challenges posed by low DNA concentrations in diagnostic applications, with the goal of enhancing detection sensitivity. One such promising approach is whole genome amplification (WGA) using the multiple displacement amplification (MDA) technique, which has demonstrated success as an enrichment method. This technique has been applied to increase the number of *B. exitiosa* genomic DNA copies extracted from the gills of various mollusc species. The amplification process was carried out using the Illustra GenomiPhi V2 Amplification Kit (GE Healthcare, Mascot, NSW, Australia), which contains Phi29 DNA polymerase, renowned for its high processivity and strand-displacement activity. The amplification was conducted at an optimal temperature of 30 °C for a duration of 90 min. After the WGA-MDA step, PCR amplification was performed to target the *actin* gene of *B. exitiosa* [19]. Although WGA-MDA is not a direct detection tool, it plays a crucial role in enhancing the genomic DNA quantity for subsequent diagnostic assays. This isothermal technique enables the amplification of *B. exitiosa* genomic material even when the initial DNA concentration is low, thereby improving diagnostic accuracy and reliability for pathogens that possess minimal genomic DNA quantities.

Another LAMP assay demonstrated a remarkable sensitivity, capable of detecting as low as 20 fg of *M. refringens*, which is 100 times more sensitive than the corresponding cPCR. The detection process was completed within 60 min [207]. These results underscore the enhanced efficiency and sensitivity of LAMP compared to cPCR, highlighting its potential as a rapid and highly sensitive diagnostic tool for detecting low-abundance pathogens. Additionally, the WOAH-recommended extraction protocol involves overnight proteinase K digestion at 50–55 °C, followed by phenol–chloroform extraction, which is time-consuming and labour-intensive [15]. To improve efficiency and accessibility, simplified extraction methods that utilise basic equipment—such as a heating block commonly available in most laboratories—should be considered as alternatives to conventional extraction kits, which are both time- and cost-intensive. Additionally, two LAMP assays were designed to identify members of the *Perkinsus* genus by targeting the conserved internal transcribed spacer 2 (ITS-2) region [177,208]. In the first LAMP assay, gill samples from living clams and suspected infected oysters were amplified using six specific LAMP primers [177]. The limit of detection for this assay was determined using a recombinant plasmid, which established a detection threshold of 10 copies within 50 min. This assay successfully detected 56 out of 60 positive samples, demonstrating greater sensitivity compared to the gold standard detection method, the RFTM, which identified 52 samples. The amplification results could be visually observed through turbidity changes or ultraviolet (UV) fluorescence [177]. The second LAMP assay was applied to tissues of Cortez and Pacific oysters [208]. This assay exhibited a detection sensitivity as low as 3.6 ng of DNA within 60 min, with amplification visualised through AGE or by adding SYBR Safe dye to the amplified products [208]. These results highlight the increasing success and widespread application of isothermal amplification techniques, such as LAMP, for the detection of a broader range of pathogens. Furthermore, LAMP assays demonstrate superior performance compared to traditional methods, offering improved sensitivity, faster detection times, and more accessible visualisation techniques. However, further studies are needed to evaluate alternative sample types—such as mucus, water, or faeces—and to assess their compatibility with existing diagnostic tools. These sample matrices must be carefully examined to ensure they do not introduce inhibitory effects that compromise assay sensitivity, thereby enabling more efficient and reliable detection in field-based settings.

Furthermore, *P. beihaiensis* could be detected using RPA. After 25 min of incubation at 37 °C, the amplification can be detected using LFD. The RPA-LFD assay could detect as few as 26 copies of the ITS region specific for *P. beihaiensis* [209]. The sensitivity of the assay was equal to the corresponding qPCR assay when tested on gill samples from oysters, *C. hongkongensis* [209]. Additionally, six primers of LAMP were designed specifically to target the *P. olseni* conserved area between 5.8S rRNA and *ITS2*. The assay was species-specific with sensitivity up to 100 fg plasmid. Kelly colour was observed in positive samples of oysters, while no colour change was observed in non-*P. olseni* samples [172]. Another LAMP assay was developed to amplify the 5.8S rDNA ITS sequences of *P. olseni*. Four primers could amplify as few as 30 copies of recombinant plasmid in one hour at 64 °C, and showed no cross-reactivity with other members of the genus *Perkinsus* [210]. Continuous research is being conducted to improve the sensitivity and specificity of isothermal methods. However, more work needs to focus on simplifying these assays so they can be commercialised and used by farmworkers for regular surveillance before pathogens spread.

**Table 2 animals-15-01664-t002:** The current isothermal assays used to detect pathogens affecting molluscs.

Pathogen	Type	Target	Sample	Duration (minutes)	Sensitivity #	In-Field	Ref.
**Virus**							
**Abalone herpesvirus (AVG)**	LAMP	DNA polymerase gene	Nerve tissues	60	100 copies/µL	No	[176]
**Abalone herpesvirus (AVG)**	RPA	ORF38	Muscle tissue	20	100 copies	No	[170]
**Acute Viral Necrobiotic Virus (AVNV)**	LAMP	-	Tissues	60	1 fg	No	[199]
**Abalone shrivelling syndrome-associated virus (AbSV)**	LAMP	ORF2	Water	60	10 copies	No	[200]
***Ostreid herpesvirus* (OsHV-1)**	LAMP	ORF 109	Tissues except for gonad and adductor muscle	60	20 copies	No	[201]
***Ostreid herpesvirus* (OsHV-1)**	LAMP	ORF 4	Tissues	60	10^3^ copies	No	[202]
***Ostreid herpesvirus* (OsHV-1)**	RPA	ORF 95	Tissues	20	207 copies	No	[204]
***Ostreid herpesvirus* (OsHV-1)**	RPA	ORF 95	Tissues	20	5 copies	No	[203]
***Ostreid herpesvirus* (OsHV-1)-SB ***	CPA	-	-	60	30 copies/µL	No	[171]
**Parasites**							
** *Bonamia exitiosa* **	MDA-WGA	Actin	Gill tissues	90	-	No	[19]
** *Bonamia exitiosa* **	LAMP	*Actin*	Gill tissues	30	50 copies/µL	No	[206]
** *Bonamia ostreae* **	LAMP	*Actin-1*	Gill tissues	30	50 copies/µL	No	[206]
***Bonamia* spp.**	LAMP	*18S*	Gill tissues	30	50 copies/µL	No	[206]
** *Marteilia refringens* **	LAMP	-		60	20 fg	No	[207]
***Perkinsus* spp.**	LAMP	Internal transcribed spacer 2 (ITS-2)	Gills/body tissues	49.8	10 copies of plasmid DNA	No	[177]
***Perkinsus* spp.**	LAMP	ITS2	Tissues	30–60	3.6–36 ng	No	[208]
** *Perkinsus beihaiensis* **	RPA	ITS	Gills	25	26 copies	No	[209]
** *Perkinsus olseni* **	LAMP	ITS 5.8S rDNA	-	60	30 copies	No	[210]
** *Perkinsus olseni* **	LAMP	Between 5.8S and ITS 2	-	-	100 fg	No	[172]

* Variant of the typical strain. # Sensitivity is calculated by reaction unless otherwise specified, loop-mediated isothermal amplification (LAMP), recombinase polymerase amplification (RPA), multiple displacement amplification (MDA), whole genomic amplification (WGA), and cross-priming isothermal amplification (CPA).

## 6. Future Improvements in the Application of Isothermal Amplification

Despite the development of several isothermal amplification assays for pathogen diagnostics in molluscs, the majority have not yet been adopted for widespread field applications. Several areas of improvement could facilitate the broader and more rapid uptake of these assays in the future. A successful field-deployable assay should feature simple, easy-to-perform sampling and extraction methods, short incubation times, the ability to process multiple samples simultaneously, and a clear, interpretable output.

A key area for improvement is the adoption of non-invasive sampling methods. All organisms shed DNA into the environment, known as environmental DNA (eDNA), and the ability to detect pathogens from environmental samples, such as water, would significantly enhance field-based disease surveillance [211]. eDNA has been successfully used to detect *P. marinus* in water samples and *Candidatus Xenohaliotis californiensis* bacteria in faecal and seawater samples [123,212]. However, further investigation is required to determine the most effective filter membranes, such as Zetapor, gauze, nylon, low-density polyethylene (LDPE), and polyvinylidene difluoride (PVDF), for capturing and eluting eDNA, considering their varying nucleic acid adsorption capacities and the specific needs for detecting different pathogens [213].

While isothermal amplification methods have demonstrated high sensitivity and specificity, further improvements can be achieved by incorporating enrichment techniques to overcome the challenges posed by low pathogen concentrations in samples. One promising approach involves the use of magnetic beads coated with anionic polymers, which have been shown to effectively capture viral pathogens in various sample matrices [214,215,216,217]. This method demonstrated high capture efficiency for viral pathogens, such as Human influenza A virus and Human immunodeficiency virus type-1 (HIV-1), with efficiencies ranging from 74% to 100%. However, lower efficiencies (5–35%) were observed for other viruses, such as Vaccinia virus and Human herpesvirus 8, when testing water samples [216].

Currently, animal tissue samples are typically used for diagnostics, with DNA extraction performed using commercial kits [19,170]. However, efficient and rapid extraction can also be achieved using simpler methods, such as adding alkaline polyethylene glycol and heating [218]. This method offers several advantages over other chemical extraction methods, including eliminating the need for neutralisation before PCR, making it suitable for viral, parasitic, and bacterial pathogens. Another simple and cost-effective lysis method, boiling tissue, was shown to yield higher DNA concentrations than five commercial extraction kits when applied to various food types, including bacon and fish eggs [219].

Most of the developed assays currently require approximately one hour to produce results, although some can deliver results in as little as 20 min (Table 2). The long incubation times, however, limit the throughput of samples and may be a constraint during disease outbreaks. Several strategies can be employed to accelerate assay results. For example, incorporating loop primers in LAMP reactions has been shown to reduce incubation times by half, potentially shortening the assay duration, which traditionally uses only four primers, if the sequence allows for the addition of loop primers [202,220,221]. Betaine has also been tested to improve specificity and sensitivity in recombinase polymerase amplification (RPA) assays. A study on RPA for hepatitis B virus detection demonstrated that the addition of 0.8 M of betaine improved test specificity and reduced the assay duration, without the need for a purification step [222]. Further testing is necessary to identify the most effective reagents for field-based detection.

Using rapid assays with portable, affordable readout methods like lateral flow devices can enhance field deployability and increase sample processing capacity [223,224]. Another promising readout method involves an electrochemical biosensor coupled with isothermal RPA for detecting OsHV-1 [204]. Ongoing research is essential to refine and optimise these detection methods, ultimately qualifying them for point-of-care use. This will be crucial for more effective disease control and prevention in aquaculture settings.

## 7. Conclusions

Mollusc production has been recognised for its environmental benefits and its role in maintaining a balanced ecosystem [2,5]. These advantages have fuelled global expansion in mollusc farming. However, this increased production has also led to a rise in disease outbreaks, which have had significant economic and socioeconomic impacts on mollusc-producing countries. Given that molluscs lack an adaptive immune system, traditional vaccine approaches are not applicable, and most diseases have no available cure [225]. Therefore, early detection is crucial to controlling the spread of infections. Most of the standard molecular detection methods mentioned in this review are not highly specific, are time-consuming, expensive, require trained personnel, and depend on a constant power supply. The absence of cell lines to isolate viral pathogens that affect molluscs, combined with the different pathogen–host immune mechanisms, emphasises the need for rapid pathogen detection, especially for viruses, which spread quickly and cause severe damage [226]. Isothermal detection assays offer a cheaper, faster, simpler alternative that does not require multiple expensive devices. However, it is rare to find an assay that has been tested under field conditions. More research is needed to develop suitable, contaminant-tolerant extraction and end-product visualisation methods that are fast and more appropriate for field use. The constraints of time, cost, and specificity demand a shift towards replacing current gold standard detection methods with more efficient, fast, and sensitive techniques, such as isothermal detection methods [169].

## Figures and Tables

**Figure 1 animals-15-01664-f001:**
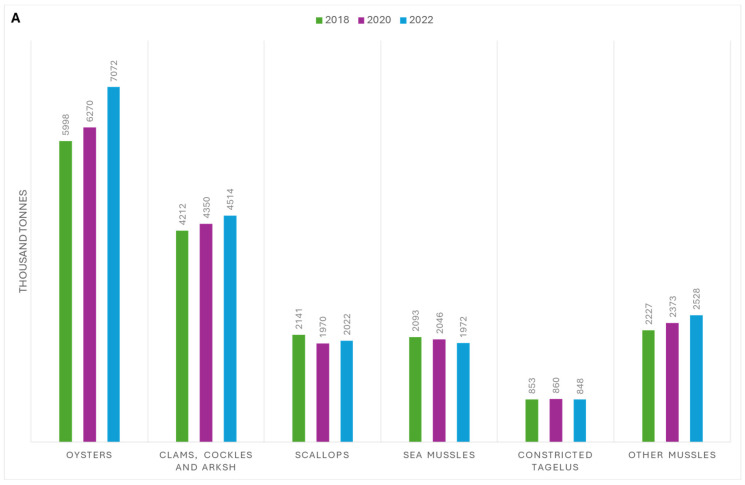
World mollusc production by type and geographical location. (**A**) The production volume of major mollusc species in 2018, 2020, and 2022. (**B**) Mollusc-producing countries worldwide and the breakdown of their production ratios [4].

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
