# Peer review of "Enhancing Biosecurity in Mollusc Aquaculture: A Review of Current Isothermal Nucleic Acid Detection Methods"

_animals, 2025, doi:10.3390/ani15111664_

Round 1
Reviewer 1 Report
Comments and Suggestions for Authors
Comments are included in the revised manuscript. I can comment that there are currently PCR and qPCR formats that can be performed in the field. However, isothermal methods are a good option.
In my opinion, the description of each pathogen is unnecessary in this document, which refers to detection methods.

Author Response
- LN 64 The absence of adaptive immunity in mollusks does not mean that they are more susceptible to diseases. The increased risk of infections in these organisms is more related to environmental quality, poor nutrition and poor management of the farm.
Thank you for your comment. We agree with the reviewer and have changed the text according.
Ln62 “The intensive expansion of aquatic farming, combined with environmental quality, poor nutrition and poor management on the farm, makes cultivated mollusc species highly susceptible to disease.”
- Not all cases of mortality are related to infections and disease.
Thank you for your comment. We concur with the reviewer's observation. However, it should be noted that infectious disease remains the leading cause of mortality, as highlighted.
- “disease symptom” is not a suitable term in aquaculture (actually in any animal farming) the correct term is signs of disease.
Thank you for your feedback. We have corrected the wording to reflect the correct term.
Ln112 “…identified as an asymptomatic carrier of the virus, harboring it without signs of disease under experimental conditions.”
- The new PCR and qPCR technologies available already include equipment that can be used in the field, so this quote is not entirely appropriate.
Thank you for your feedback. We agree there are qPCR and PCR technologies that are proposed to field deployable however do still have drawbacks. We have additional paragraph.
Ln435 “Field-deployable qPCR instruments are emerging, particularly for biodefence (Ozanich et al., 2017), but their use is still nascent. The genesig q16 (Primerdesign UK) is portable and field-operable, validated for salmonid alphavirus detection (Stagg et al., 2018), with over 500 assays claimed (https://www.genesig.com/products/9696-genesig-q16-real-time-pcr-instrument). Similarly, BioFire FilmArray (BioFire Diagnostics; https://www.biofiredx.com/) is a compact, portable qPCR system for anthrax and other biodefence threats, functioning as a closed system for nucleic acid extraction and amplification, albeit processing one sample at a time. In addition, both the Genesig and BioFire instruments require alternating current (AC) power to operate, which could limit their usefulness in the field.”
- Currently there are very efficient commercial extraction kits in field format based on silica adsorption.
Thank you for the feedback. We acknowledge that silica adsorption-based DNA extraction kits, while highly efficient, involve multiple steps such as centrifugation and sequential buffer additions. These requirements can be challenging to implement in low-resource settings or remote aquaculture farms, particularly in developing countries where access to such kits and necessary equipment may be limited. We stated that extraction is simple as heating with buffer.
- I do not agree with this comment, because what would be the difference between delivering a result in 30 minutes or an hour compared to delivering it in 4 hours? I do not believe that the difference is significant for the decision of the actions to be carried out in an aquaculture farm.
Thank you for your comment. We have rewritten this section
Ln779 “Most of the developed assays currently require approximately one hour to produce results, although some can deliver results in as little as 20 minutes (Table 2). The long incubation times, however, limit the throughput of samples and which may be a constraint during disease outbreaks.”
Q7. Comments are included in the revised manuscript. I can comment that there are currently PCR and qPCR formats that can be performed in the field. However, isothermal methods are a good option.
Answered in Q4.
Q8. In my opinion, the description of each pathogen is unnecessary in this document, which refers to detection methods.
Thank you for your feedback. While the primary focus of the document is on detection methods, we believe that including brief descriptions of each pathogen is important for context. This background helps readers—particularly those less familiar with the specific pathogens—understand why certain detection methods are appropriate or necessary. It also aids in comparing diagnostic approaches across pathogens with differing biology, transmission dynamics, and field relevance.
Reviewer 2 Report
Comments and Suggestions for Authors
The review article on ‘Advances in Isothermal Nucleic Acid Amplification Technologies for Detecting pathogens of commercial molluscs’ largely failed to justify the title
- Instead the authors focus on molluscan production, diseases and its diagnostic methods rather than reviewing the advancement in the isothermal amplification methods
- The authors should have focussed more on
- Available isothermal diagnostic methods including Helicase- dependent amplification and others
- Advantages of isothermal amplification over other methods such as PCR for detecting molluscan pathogens
- End point detection methods in isothermal amplification methods such as precipitation based, colorimetric based, and LFA.
- Recent advances in isothermal amplification like electrochemical and biosensor, smart-phone based and field deployable devices
- Limitations and challenges of isothermal amplification methods in sensitivity, specificity, multiplexing.
Author Response
Q1. The review article on ‘Advances in Isothermal Nucleic Acid Amplification Technologies for Detecting pathogens of commercial molluscs’ largely failed to justify the title. Instead the authors focus on molluscan production, diseases and its diagnostic methods rather than reviewing the advancement in the isothermal amplification methods.
Thank you for your feedback. We have changed the title to be more suitable for our review. The new title is “Enhancing Biosecurity in Mollusc Aquaculture: A Review of Current Isothermal Nucleic Acid Detection Methods”
Q2. The authors should have focussed more on
- Available isothermal diagnostic methods including Helicase- dependent amplification and others
- Advantages of isothermal amplification over other methods such as PCR for detecting molluscan pathogens
- End point detection methods in isothermal amplification methods such as precipitation based, colorimetric based, and LFA.
- Recent advances in isothermal amplification like electrochemical and biosensor, smart-phone based and field deployable devices
- Limitations and challenges of isothermal amplification methods in sensitivity, specificity, multiplexing.
Thank you for your feedback. We have revised the title to more accurately reflect the scope of our review. Our intention was not to cover all aspects of isothermal amplification, detection methods, and multiplexing. There are numerous recent reviews that address these topics comprehensively. Instead, our review focuses specifically on the application of isothermal detection in mollusc aquaculture, and we have mentioned several relevant points throughout the manuscript.
Reviewer 3 Report
Comments and Suggestions for Authors
Dear Authors
The document presents an exhaustive analysis of contemporary isothermal nucleic acid amplification methodologies—namely LAMP (Loop-mediated Isothermal Amplification), RPA (Recombinase Polymerase Amplification), CPA (Cross-Priming Amplification), and MDA (Multiple Displacement Amplification)—alongside their utilization in identifying viral and parasitic pathogens impacting commercially cultivated molluscs. It also comprises a comprehensive table describing current isothermal tests and delineates future prospects for the implementation of these technologies.
The publication should address the quality and validity of the relevant assays, specifically their adherence to the standards established in the Principles and Methods of validity of Diagnostic Assays for Infectious Diseases as delineated by the WOAH. This is a crucial factor for the advancement and practical application of these technologies. Furthermore, it is recommended to rectify current deficiencies or baseline criteria for the implementation of these diagnostic instruments in the field, particularly at the farm site level. Regards,
Author Response
- The document presents an exhaustive analysis of contemporary isothermal nucleic acid amplification methodologies—namely LAMP (Loop-mediated Isothermal Amplification), RPA (Recombinase Polymerase Amplification), CPA (Cross-Priming Amplification), and MDA (Multiple Displacement Amplification)—alongside their utilization in identifying viral and parasitic pathogens impacting commercially cultivated molluscs. It also comprises a comprehensive table describing current isothermal tests and delineates future prospects for the implementation of these technologies.
Q1. The publication should address the quality and validity of the relevant assays, specifically their adherence to the standards established in the Principles and Methods of validity of Diagnostic Assays for Infectious Diseases as delineated by the WOAH. This is a crucial factor for the advancement and practical application of these technologies.
Thank you for your feedback, We added few sentences to each of the isothermal detection assays mentioned in section 4 and section 5 to compare it with the current methods listed in the WOAH. However some diseases are not listed in WOAH therefore we evaluated them according to other published assays.
L575 Although both isothermal assays demonstrate high sensitivity, the TaqMan PCR assay recommended by WOAH remains superior, with a detection limit as low as 30 copies per reaction. To enhance the performance of current isothermal detection methods, optimization of sampling and nucleic acid extraction protocols is necessary. At present, both assays rely on abalone tissue processed using commercial extraction kits, which are costly, require trained personnel, and are impractical for field use [201, 202]. Therefore, in-field validation of both isothermal techniques under realistic aquaculture conditions is essential to support their future application in routine diagnostics.
L591 Although the assay can sensitively detect low levels of the virus in animal tissue, its dependence on invasive sampling methods limits its practicality for field application. To enhance its suitability for in-field diagnostics, non-invasive sampling approaches should be developed and thoroughly evaluated for the detection of AVNV.
L597 The detection limit of this assay is comparable to that of corresponding qPCR; however, both currently require invasive tissue sampling through animal dissection.
L627 Both the developed isothermal assays and the conventional cPCR and qPCR methods recommended by the WOAH currently rely on combined gill and mantle tissues as the sample source [201]. For more efficient diagnostics and routine surveillance, it is recommended to evaluate simple, non-invasive sampling techniques—such as swabbing—as alternative sources of DNA for both isothermal and PCR-based assays. Implementing such methods would improve field applicability, particularly in farm environments and resource-limited laboratories, while also supporting mollusc health monitoring and minimizing the need for unnecessary animal sacrifice.
L710 While isothermal detection methods offer faster results and require less technical expertise compared to traditional approaches such as tissue imprints, histology, and the WOAH-recommended PCR assays, all of these methods still depend on invasive sampling and complex DNA extraction protocols. To facilitate regular surveillance under field conditions, there is a critical need to develop non-lethal sampling techniques and simplified extraction methods.
L737 Additionally, the WOAH-recommended extraction protocol involves overnight proteinase K digestion at 50–55 °C followed by phenol-chloroform extraction, which is time-consuming and labor-intensive [201]. To improve efficiency and accessibility, simplified extraction methods that utilize basic equipment—such as a heating block commonly available in most laboratories—should be considered as alternatives to conventional extraction kits, which are both time- and cost-intensive.
L759 However, further studies are needed to evaluate alternative sample types—such as mucus, water, or feces—and to assess their compatibility with existing diagnostic tools. These sample matrices must be carefully examined to ensure they do not introduce inhibitory effects that compromise assay sensitivity, thereby enabling more efficient and reliable detection in field-based settings.
Q2. Furthermore, it is recommended to rectify current deficiencies or baseline criteria for the implementation of these diagnostic instruments in the field, particularly at the farm site level.
Thank you for your feedback. The future improvements section includes recommended methods to make the current isothermal detection methods more suitable for field-deployability however according to your feedback, We added more information about the limitations of the current assays following each of the isothermal detection assays as illustrated in Q1.